# Is It Possible to Become a Nurse in a Refugee Camp?

**DOI:** 10.3390/ijerph16183414

**Published:** 2019-09-14

**Authors:** Olga María López-Entrambasaguas, Jose Manuel Martínez-Linares, Manuel Linares-Abad, María José Calero-García

**Affiliations:** Faculty of Health Sciences, University of Jaén, Campus Las Lagunillas s/n, Edificio B3. CP, 23017 Jaén, Spain; omlopez@ujaen.es (O.M.L.-E.); mlinares@ujaen.es (M.L.-A.); mjcalero@ujaen.es (M.J.C.-G.)

**Keywords:** nursing education, international cooperation, nursing school, refugee camps, global health

## Abstract

The history of the Western Sahara has been marked by several events that have contributed to the protracted refugee situation in which the Sahrawi people have found themselves since 1975: the Spanish colonization and the subsequent decolonization process, the armed struggles between the indigenous population and the states of Morocco and Mauritania to occupy Western Saharan territory, assassinations and repression of the Sahrawi population, and the economic interests of external agents with regards to mineral resources. Twenty-five years ago, in the hostile environment of the Sahrawi refugee camps, a nursing school was founded. Essentially depending on foreign aid, this school has been responsible for training nursing professionals to meet the healthcare needs of the population. The aim of this paper is to provide an approach to the origin and evolution of nursing education for the Sahrawi refugee camps. The Sahrawi are the only refugee camps in the world to host such nursing schools.

## 1. Introduction

The Western Sahara was a Spanish colony [1] until Morocco entered the territory in an invasion [2] that was dubbed the “Green March”, in November 1975. Spain had already committed itself to begin decolonization, agreeing to hold a referendum on independence, but the Green March caught Spain at a difficult moment; with the dictator Francisco Franco on his deathbed, Spain chose to abandon the territory in February 1976.

In conformity with the Madrid Accords, Spain withdrew from Sahrawi territory, leaving it in the hands of Morocco and Mauritania, which provoked a mass exodus of Saharawi people towards Algeria [2]. At the same time, the Polisario Front (Popular Front for the Liberation of Saguia el Hamra and Río de Oro), which had turned into the Sahrawi liberation movement, proclaimed the Sahrawi Arab Democratic Republic (SADR) and waged a war of liberation against its invaders (Morocco and Mauritania). The war continued until 1991, when both parties signed a ceasefire under the auspices of the United Nations, which took responsibility for finding a peaceful solution to the Sahrawi conflict and established the United Nations Mission for the Referendum in Western Sahara (MINURSO in Spanish) [3]. The call has been delayed for various reasons, which means that the referendum has not yet taken place [4].

The invasion of Sahrawi territory forced tens of thousands of people to emigrate inland to Algeria, settling near the city of Tindouf in five refugee camps (or five wilayas, a term used in the Arabic dialect of the Sahrawi people, Hasaniyya): Dakhla, Smara, Laayoune, Awsard, Boudjour, and the administrative center, Rabouni (Figure 1).

About 185,000 people have lived here for the last 43 years, lacking proper health facilities and trained health professionals, surviving in precarious conditions on humanitarian aid [6,7] and hoping to return to their homeland (Figure 2).

This paper is a result of the project of international development cooperation and its contribution to the improvement of the Sahrawi population’s quality of life through institutionally strengthening the University of Tifariti [8]. It was funded by the Andalusian Agency for International Development Cooperation (Spain), and carried out by the University of Jaén during 2014–2018 at the Sahrawi refugee camps. This project analyzed the situation at the Ahmed Abdel-Fatah School of Nursing with the aim of contributing to its improvement.

## 2. Literature Review

It was not until the late 19th century that the first attempts of occupation of the Sahrawi territory were made. The Spanish presence in the Western Sahara started in 1884, due to interests in their fishing resources and their geostrategic location close to the Canary Islands [9]. Upon approval of Resolution 1514 of the United Nations [10], the international community pressured Spain to decolonize the Western Sahara.

In 1973, Spain announced the purpose to hold a self-determination referendum and leave the Western Sahara. Morocco and Mauritania, for their part, opposed such a referendum and reclaimed sovereignty over the Sahrawi territory by alleging historical rights. Thus, Morocco commenced the so-called “Green March” in 1975, a pacific invasion headed by 350,000 unarmed Moroccans. The weakened Spanish government, for its part, left the Sahrawi territory in the hands of Morocco and Mauritania with the execution of the Madrid Accords by the end of 1975. The withdrawal of the Spanish troops concluded on 27 February 1976. That is when the Polisario Front proclaimed the SADR and a war broke out. Mauritania withdrew from the fights and the Moroccan army occupied the territory thanks to the support of France and the United States [11]. This provoked the exodus of thousands of Sahrawi people from their cities of origin into the desert, crossing the Algerian borders and settling down in the current Sahrawi refugee camps of Tindouf, managed by the Polisario Front [12].

This area of the desert is one of the most inhospitable places on Earth due to its arid weather, with extremely high temperatures and major thermal changes. The people living in the Sahrawi refugee camps do not have their own resources and subsist thanks to humanitarian and international development cooperation. Their economic system is virtually non-existent, although it grows slowly. There are small shops which distribute basic goods to supplement those commodities provided by humanitarian aid. These basic goods are essentially food, clothes, footwear and housewares. Foreign cooperation becomes fundamental to favor and foster the development of production activity which serves as a base for economic growth, with the aim of achieving a greater independence from cooperation [6].

This situation, which has been ongoing for 43 years now, has caused two generations of Sahrawi people to be born and live in exile, with the consequent limitations in future choices. Sixty percent of the population of the Sahrawi refugee camps is estimated to be below 30 years old [13]. Even though Sahrawi youngsters could thrive thanks to the education they are provided with, the lack of resources and the precariousness prevents them from being able to apply such knowledge, which has provoked an increase in truancy and crime. In view of the above, the situation requires political agreement among the parties involved in order to unblock negotiations [14].

Amid the deadlock of such political agreement, Sahrawi refugees’ harsh living conditions are perpetuated, which allows the government of the SADR to receive international aid from the United Nations High Commissioner for Refugees (UNHCR), Red Cross, Médecins Sans Frontières, Doctors of the World, and several Non-Governmental Development Organization (NGDO); and develop the infrastructures necessary to provide basic healthcare. For such a purpose, it was necessary to provide a nursing education program with young people including refugee youth, who would later become professional nurses and help the refugees. The education program could not only fill the career and personal needs of the young people but also meet the health care needs of the refugees. This whole process includes establishing the first nursing school in any refugee camp across the world. Nevertheless, the extant literature has not documented this unique context yet.

There are no articles in the scientific literature which document the creation of this nursing school that, as mentioned, is the only nursing school existing in a refugee camp. The only references to this nursing school can be found on the websites of the different organizations that helped build and create it [15,16,17,18,19,20]. Thus, the present article is intended to fill this gap, as it is the first article to document the existence of the school.

## 3. Material and Methods

Data collection was conducted through personal interviews of people who had been a part of the managing or teaching staff of the Ahmed Abdel-Fatah School of Nursing (Table 1) since foundation and who were still living in the refugee camps, even when they were no longer a part of such a center. Nine interviews were conducted, each of which took approximately 45–60 min. However, the interviews were not recorded on audio or video because the interviewees did not permit our recording. Field notes were taken and the information retrieved was used to elaborate a report [21], which was one of the results of the above-mentioned international cooperation project. Tis report was returned to the interviewees so that they could correct any transcription mistakes. Fortunately, no mistakes were detected and no interview had to be repeated.

No participant refused to be interviewed, having been previously informed about the goal of the interview and signing the corresponding informed consent forms. Interviewees’ sociodemographic characteristics are as follows:

In the same way, information was obtained through consultation of filed documents and activity logs of the educational center to verify the data collected during the interviews. The obtained documents provided supplemental information, which can enrich the interview data. This consultation was conducted in the presence of some members of the management staff of the center after obtaining the corresponding informed consent to do so. No pictures of the aforementioned documents were taken.

## 4. Results

The Sahrawi are the only refugee camps where one can study and become a nurse. Considering the perpetuity of such camps and the lack of healthcare professionals to serve the refugees population, building a nursing school inside the refugee camps is a priority for the Sahrawi government based on partnerships with a range of international organizations whose mission is to help refugees. The following lines detail information on the context and lifestyle of the Sahrawi refugees, as well as the history and evolution of the nursing school which developed in this environment.

### 4.1. Geopolitical and Economic Context of the Sahrawi Refugee Camps

The Sahrawi people are currently divided into the territories occupied by the Moroccan military, the “free zone” governed by the Polisario Front and the refugee camps of Tindouf (Algeria). They have very meager resources of their own, and have to subsist essentially on humanitarian aid and international development cooperation. Western Sahara is on the list of the sixteen Non-Self-Governing Territories under the supervision of the United Nations Special Committee of Decolonization. At the moment, the territory is divided by a wall about 2720 km long that separates Morocco’s “occupied territories” (approximately 75% of the region) to the west of the wall, from the eastern area, which is governed by the Polisario Front and proclaimed as the Sahrawi Arab Democratic Republic (SADR), and is denominated as the “free zone” by the Sahrawi people [11].

The economic system of the Sahrawi refugee camps is practically non-existent. External cooperation projects for the development of manufactured goods and craftwork markets are necessary in order to support and promote the development of centers of production as a basis for economic development, thus enabling them to fulfill their basic needs and affording them greater independence from foreign aid [12].

### 4.2. Education in the Sahrawi Camps

To properly understand the context, it must be borne in mind that formal education in Sahrawi refugee camps is considerably different from that of any other country. Since resources are so scarce, everything requires a great deal of effort and international cooperation, such as those provided by professionals [22], and is carried out with the conviction that education is essential in order to provide the population with a decent future.

For the Saharawi people, education—as any other activity—is geared towards achieving two main goals: making students aware that their studies help the Sahrawi state to progress, and that the qualification they obtain is intended to serve the community. From these two goals, it can be understood that everyone in Sahrawi society works collaboratively to help their education system to flourish [23].

### 4.3. Ahmed Abdel-Fatah National School of Nursing: Past, Present, and Support from International Cooperation

Due to the lack of nursing professionals to provide refugees with basic healthcare and fully taking into account the above two goals, the SADR government, thanks to international aid, created the Ahmed Abdel-Fatah National School of Nursing.

The public health service of the SADR was initially staffed by those few nurses and assistants that had been trained at Spanish schools during the Spanish occupation. Despite their precarious, difficult conditions with regards to the paucity of staff and of medical materials and equipment, this handful of staff managed to look after the health of the citizenry. Nevertheless, the population’s needs were far from being fulfilled.

Once the medical institution was up and running, a nursing school was built in the refugee camps, which opened its doors on 12 October 1992, to mark the twenty-seventh anniversary of national unity. The facilities of an old hospital on the outskirts of the wilaya of Smara were restored for use as a small teaching center and boarding house, since the distance to the nearest wilaya made it impossible for teachers and students to travel there daily.

It was the Ministry of Public Health that first had the idea of creating the school, in line with the objectives drawn up by the Sahrawi state, which were aimed at achieving self-sufficiency. The goal was to remedy the dearth of assistants, nurses and specialized nurses in various local, regional and national institutions and the Sahrawi health organization in general.

The Ahmed Abdel-Fatah School of Nursing gradually expanded to become a national institution offering work and educational competences, and it now collaborates with healthcare institutions in order to help improve them.

Training began—using the meager resources at their disposal—in stages that usually lasted no more than six months. For the first two years, the school only trained nursing assistants; recent graduates were immediately employed as teaching staff, even though they lacked training in teaching and pedagogy, in order to keep the program in operation.

While working with the French NGDO, Enfants Refugiés du Monde (Refugee Children of the World), it became clear that a new teaching style was needed. Offering a three-year general nursing degree, as well as a special one-year program, meant that the school was able to admit and train greater numbers of students in successive classes.

In 2005, the school began a new phase after in-depth discussions on educational matters between NGDOs and the school’s management; an agreement was reached on the system for assessing the nursing students’ training, in practical as well as theoretical fields.

A lecture room building was constructed in 2010, financed by Grupo CIVICA, a construction company from Alicante, Spain, with support from the University of Alicante in terms of materials and other contributions. The new building allowed the school to offer a greater number of places to students, as well as separate the teaching area from the residential and rest area. This led to a substantial improvement in the quality of the teaching and training, as well as in the living conditions of the teaching staff and students (Figure 3).

This building contains four classrooms, a library, a laboratory, a computer lab, and a conference hall. The library is furnished and has a variety of nursing books, although they are not catalogued or ordered. The Asociación de Amigos de la RASD (Friends of the SADR Association), in Vitoria-Gasteiz, Spain, has developed literature on various subjects related to nursing for teachers and students in Arabic and Spanish. Likewise, this association is currently developing a new, extended bibliographic catalogue.

To increase the training on offer to include certain specialties, with the collaboration of Doctors of the World, another building was constructed for training midwifery professionals, which was inaugurated at the beginning of the 2011–2012 academic year. This second building consists of three classrooms, a laboratory, and a computer lab, as well as a greater number of places in the boarding house for new students and teachers of this specialty, with their respective equipment [24]. The school currently has 14 teachers and 13 administrative and service personnel.

A list of non-governmental development organizations that have collaborated with the Sahrawi School of Nursing since its inception can be found in Table 2. They have supported the school, in terms of both infrastructure and pedagogical activities.

### 4.4. Current Academic Curriculum of the Nursing School

With the support of the Ministry of Public Health, the school developed an academic curriculum that includes 18 subjects plus practical classes distributed throughout the three years of the course. During the first year, in addition to performing first aid practices in the laboratory, the following subjects are taught: Fundamentals of Nursing, Anatomy and Physiology, Microbiology and Epidemiology, Nutrition, Professional Ethics, Calculation, and Spanish Language. All of these are annual except for Professional Ethics, which lasts for one semester.

During the second year, the following subjects are taught: Anatomy, Medical-Surgical Pathology, Maternal-Child Nursing, Psychology, Spanish Language, Calculation, Pharmacology, Basics of Nursing, Community Nursing, and Information Technology (IT). As in the previous course, all of these are annual except for Community Nursing, which lasts for one semester. In the second year, students also start their internships at the national hospital and regional hospitals, alternating 15-day periods of theoretical and practical classes. Internships represent 40% of the course load, and students spend different lengths of time in each of the hospital departments depending on the number of students undertaking the internships.

During the third year, the subjects are as follows: Pharmacology II, Medical-Surgical Pathology II, Management, Healthcare Programs PISIS (Sahrawi Comprehensive Child Health Program) and Primary Care Guide, Psychiatry and Psychology, Research Methodology, Spanish Language, Calculation, IT, Basics of Nursing, and National History.

In the third year, all subjects are taught during the first semester, since the second semester is devoted to practical work in the regional hospitals (Figure 4), the National Hospital in Rabuni and the dispensaries in the wilayas. Internships represent 60% of the course load. During the third year, students must complete a degree thesis to be submitted and presented at the end of the course.

Specialties in midwifery [6] and pediatrics nursing are offered since 2002 and 2012, respectively. During the 2011–2012 academic years, in coordination with the Autonomous University of Madrid, the Ministry of Public Health decided to develop a nursing consultant specialty; however, the course was only taught that year and was thereafter discontinued [25].

### 4.5. Undergraduates and Graduates of the Nursing School, 1992–2018

Since its inauguration on 12 October 1992, the school has always tried to help the health sector by offering courses in various specialties relating to the population’s needs. Two classes of Nursing Assistants, 22 of General Nursing, 11 of Midwifery and 4 of Pediatric Nursing have graduated from this school.

Over the last 25 years (1992–2018), 301 nurses, 68 midwives and 21 pediatric nurses have been trained, making a total of 390 graduates of this school; these have been placed by the Ministry of Public Health among the national hospital, regional hospitals and local dispensaries, as required. Table 3 shows the number of students who enrolled and those who graduated in Nursing according to each year of intake.

One of the main problems observed during the interviews and the document analysis used to carry out the fieldwork was the fact that many women who start their studies at the school do not complete their degree, as their families oblige them to get married or to take on the burden of family care. As a result, there is a high dropout rate in the second year: less than 50% of those who enrolled in general nursing reach the third year at the school, while for the midwifery specialty there are 20% fewer remaining students. As a consequence, more women than men enroll in the first year, but more men than women complete their studies and enter the labor market.

Once the students have completed their training and received their degree certificate (Figure 5), the school sends a list of the graduates to the Ministry of Public Health, which places them in hospitals as required.

### 4.6. What Has Been Reached and What Is yet to Come

Thanks to the wealth of experience that this institution has gained in the 25 years since it opened, tangible results have been achieved for the health sector and the Sahrawi society in general, among which we would like to highlight:Collaborating in the improvement of health services through the support of qualified staff.Allowing for the admission of more students after the school’s expansion.Preparing textbooks and teaching models according to the needs and resources at hand.Improving the training and teaching of students.Offering retraining and further education in practical and theoretical skills, for both teachers and students.

Nevertheless, the scarcity of available resources means that conditions for the teaching–learning process for both teachers and students are not ideal. Therefore, in the coming years, the school aims to create the conditions needed to meet the needs of Sahrawi society. The actions that will help these goals to be achieved are as follows:-Seeking funding for future training courses, such as Nursing Administration and Management, Emergency Nursing and retraining for graduates.-Coordinating with cooperation agents to ensure the pedagogical training of the school’s teachers.-Consistently guaranteeing better living conditions for students and teachers, as well as electrical and Internet supply during training periods.-To construct a building for the Pediatric Nursing specialty.

## 5. Discussion

The greatest population exodus ever registered is currently happening. According to the data of the UNHCR, there are currently 70.8 million people worldwide who have been forced out of their territories. Of these, 25.9 million are refugees, half under 18 years old. Education provides refugee children and adolescents with a place where they can feel safe and become individuals who contribute to rebuild their countries upon reestablishment of certain conditions which allow a safe return [26].

While, in 2016, 91% of children worldwide were schooled in primary education, only 61% of refugee children were schooled, which means that only 23% of refugee adolescents may take secondary studies, compared to 84% non-refugees [27]. The situation increasingly worsens when it comes to higher education: 36% of young non-refugees are schooled in higher education, while only 1% of refugees can gain access to it [27]. Thus, UNHCR declares that education must be a part of any response given in a situation of refugee crisis and that the teaching staff must have as many materials and as much attendance and payment as they deserve.

The literature does not cover the existence of any center where people can take higher education studies in a refugee camp, except for Ahmed Abdel-Fatah School of Nursing in the Sahrawi refugee camps, which is integrated in the University of Tifariti, founded in 2012 [28], and which has the institutional support of different universities. This can be considered the only university to have been founded in a refugee camp.

One of the characteristics that all refugee camps have in common is their provisionality, that is, they are temporary solutions and there is not to create a city-camp forever [29]. However, some of the currently existing camps have a history of several decades, as is the case of Jabalia in Gaza, Katumba in Tanzania or Panian in Pakistan. The perpetuation of this situation has given rise to the creation of institutions and infrastructures in some of them to provide for the demands of the population. Nevertheless, none of the above mentioned camps host as many people as Tindouf Sahrawi refugee camp. In addition, the Western Sahara is still a non-self-governing territory as per the United Nations. Due to all this, the SADR now considers education and teaching to be two essential points in liberation, progress and change into better life conditions [28].

As Malala Yousafzai said when she was awarded with the Nobel Peace Prize, “One child, one teacher, one book, one pen can change the world” (10 December 2014). The geographical environment where this School of Nursing lies and the lack of resources available enhance chronicity of its dependence on external aid and cooperation. Thus, provision of school and laboratory practice materials, modernization of the bibliography used in teaching and learning, training in use of information and communication technologies applied to teaching and update of teaching staff as per contents and new teaching strategies were some of the detected weaknesses, the solution for which may be enhanced by the start-up of both cooperation and aid projects and future research.

## 6. Conclusions

The invasion of Sahrawi territory forced the emigration of thousands of people. They have to subsist on humanitarian aid and international development cooperation. In Sahrawi refugee camps education try to provide the population a decent future. Is possible to become a nurse in the Sahrawi refugee camp. The Ahmed Abdel-Fatah National School of Nursing have trained 301 nurses since 1992. The school developed a three-years academic curriculum, including subjects and practical classes. Also, it aim is to create the conditions needed in order to meet the needs of Sahrawi society. 

## Figures and Tables

**Figure 1 ijerph-16-03414-f001:**
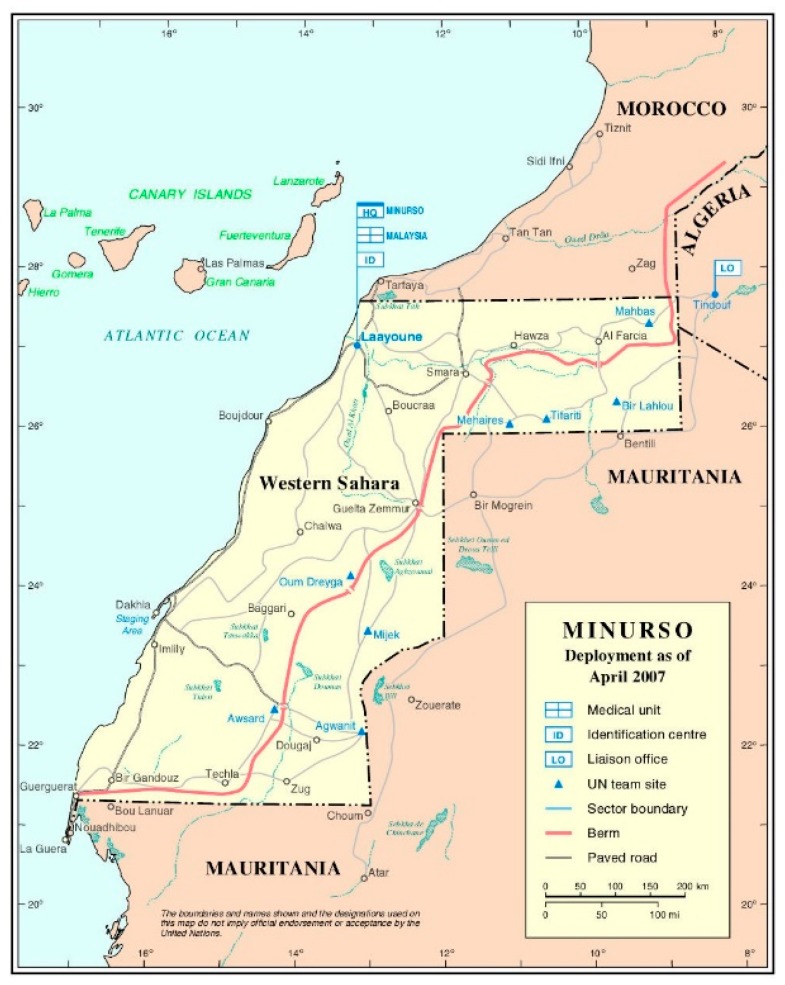
Sahrawi refugee camps area and detail of the wilayas. Source: United Nations. Cartographic section maps, 2007 [5].

**Figure 2 ijerph-16-03414-f002:**
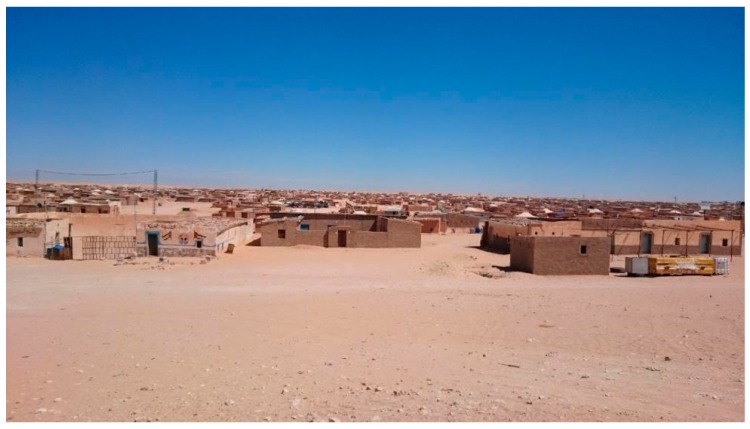
One of the Sahrawi refugee camp. Source: author’s own photo.

**Figure 3 ijerph-16-03414-f003:**
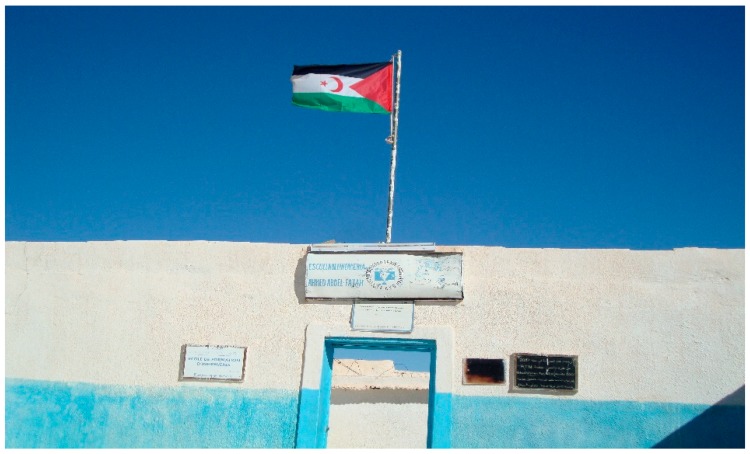
Abdel-Fatah School of Nursing. Source: Author’s own photo.

**Figure 4 ijerph-16-03414-f004:**
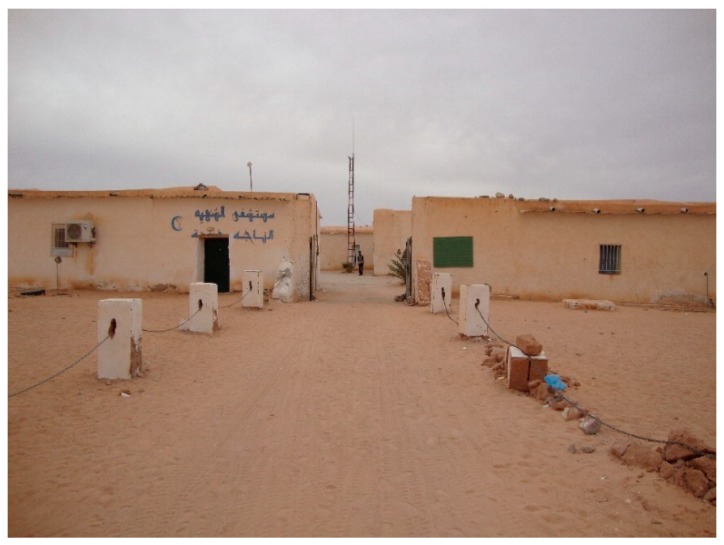
Regional hospital at the wilaya Awsard. Source: author’s own photo.

**Figure 5 ijerph-16-03414-f005:**
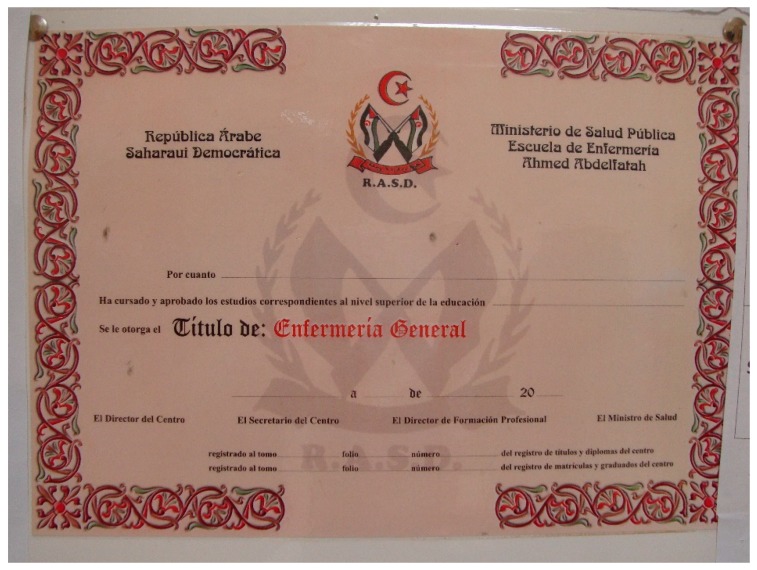
Official certificate of General Nursing. Source: author’s own photo.

**Table 1 ijerph-16-03414-t001:** The sociodemographic data of interviewees.

Interview Code	Post	Sex	Age	Experience (Years)	Nationality
I1	Head	Male	68	36	Sahrawi
I2	Professor	Male	28	5	Sahrawi
I3	Professor	Female	25	2	Sahrawi
I4	Professor	Female	42	19	Sahrawi
I5	Professor	Female	26	2	Sahrawi
I6	Professor	Male	35	13	Sahrawi
I7	Professor	Male	34	10	Sahrawi
I8	Head of studies	Male	56	24	Sahrawi
I9	Secretary	Female	58	21	Sahrawi

Source: prepared by the author.

**Table 2 ijerph-16-03414-t002:** List of cooperation agents that have collaborated with the Ahmed Abdel-Fatah School of Nursing.

NGDO/Association	Funding Body	Action Undertaken
Enfants Refugiés du Monde [14]	United Nations High Commission for Refugees City of Rezë City Hall of Pays de la Loire	Improving the school’s infrastructure Theoretical and practical training
Asociación de Amigos y Amigas del Pueblo Saharaui de Vitoria (Friends of the Sahrawi People Association of Vitoria, Spain) [15]	Council of Vitoria City Hall of Vitoria	Improving the school’s infrastructure Theoretical and practical training
Asociación DAJLA de Amigos y Amigas del Pueblo Saharaui de Alicante (DAJLA Friends of the Sahrawi People Association of Alicante) [16]	City Hall of San Vicente University of Alicante	Improving the school’s infrastructure Theoretical and practical training
Doctors of the World (Spain) [17]	Spanish International Development Cooperation Agency	Improving the school’s infrastructure
Asociación Alianza Sáhara-Salud (Sahara Health Alliance Association, Spain) [18]	Autonomous University of Madrid	Theoretical and practical training for Nursing Consultants
University of Jaén [19]	University of Jaén Andalusian International Development Cooperation Agency	Improving pedagogical performance

Source: prepared by the author.

**Table 3 ijerph-16-03414-t003:** Admissions and graduates of the General Nursing qualification of the Ahmed Abdel-Fatah School of Nursing, by class.

Class	Enrolled	Graduates
1992–1995	32	07
1995–1998	45	17
1996–1999	31	08
1997–2000	25	12
1998–2001	30	15
1999–2002	26	09
2000–2003	32	13
2001–2004	35	11
2002–2005	29	15
2003–2006	35	13
2004–2007	37	13
2005–2008	35	15
2006–2009	38	14
2007–2010	40	19
2008–2011	32	10
2009–2012	38	14
2010–2013	35	09
2011–2014	31	18
2012–2015	35	17
2013–2016	44	16
2014–2017	45	17
2015–2018	42	19
Total	737	301

Source: prepared by the author.

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
