# Peer review of "Is It Possible to Become a Nurse in a Refugee Camp?"

_ijerph, 2019, doi:10.3390/ijerph16183414_

Round 1

Reviewer 1 Report

The manuscript entitled “Is it possible to become a nurse in a refugee camp?” is informative. However, the authors need to include “method” section that describes data collection and analysis process. For example, in page 7, the authors seemed to conduct interviews and content analysis on certain forms of documents. If so, the authors must indicate how many interviewees were involved, what are the demographic characteristics of the interviewees (e.g., gender and race/ethnic backgrounds), how long each interview took, etc. Also, regarding the document analysis, the authors need to indicate where the authors located the documents, what were the selection criteria of the documents, and the way of conducting content analysis on the selected documents. After “Method” section, the authors need to create “Results” section which includes the contents from Geopolitical and economic contexts of the Sahrawi refugee camp (page 3) to the end of this manuscript (page 9). Most importantly, the “Results” section should answer the research question, “Is it possible to become a nurse in a refugee camp?” (if this is the research question for this study). Following Results or Finding section, the authors should create Discussion section that provides implications for practices and future research based on the authors’ findings. Moreover, right after the introduction section, there must be the section for “literature review” which provides an overview of Sahrawi refugee. Then, the authors should clarify how the authors’ study can fill the gap in the literature and what are the research questions for this study. Besides, the authors may need to consider the following points to enhance readability in the current form of the manuscript.

·       Abstract: “The only nursing school that exists worldwide in a refugee camp” should be edited. This is not a sentence, but must be a sentence. For example, “Sahrawi refugee camp is only the refugee camp worldwide where has nursing schools.”

·       In page 2, please revisit and reword the sentence beginning “This paper is a result of the project of international development cooperation Contribution to the…” This sentence is too long, and it is not clear why the word “Contribution” began the upper case, “C.”

·       In page 3, under the section Geopolitical and economic context of the Sahrawi refugee camp, please clarify the meaning of “external cooperation.” It would be helpful to add some examples related to “external cooperation.”

·       In Page 3, there should be a connection between “Education in the Sahrawi camps” and “Ahmed Abdel-Fatah National School of Nursing:...” The nursing school might be established based on the two main goals that the authors described in the second paragraph of the section “Education in the Sahrawi camps.” If so, please add how the two main goals can be applied to the mission of the nursing schools.

Author Response

Cover letter – Reviewer 1.

Thank you very much for the suggested modifications for the paper Is it possible to become a nurse in a refugee camp? Having them already done, all the authors believe that the article has improved.

First of all, we reconfirm the author list and the corresponding affiliation of them.

Every modification done is described below. In the paper you can find them in green colour:

The manuscript entitled “Is it possible to become a nurse in a refugee camp?” is informative. However, the authors need to include “method” section that describes data collection and analysis process. For example, in page 7, the authors seemed to conduct interviews and content analysis on certain forms of documents. If so, the authors must indicate how many interviewees were involved, what are the demographic characteristics of the interviewees (e.g., gender and race/ethnic backgrounds), how long each interview took, etc. Also, regarding the document analysis, the authors need to indicate where the authors located the documents, what were the selection criteria of the documents, and the way of conducting content analysis on the selected documents.

Thank you very much for your suggestions. We have added a “Material and methods” section in page 3, including how the interviews and consult of documents were carried out (lines 114-136) and a table with sociodemographic data of interviewees (Table 1).

After “Method” section, the authors need to create “Results” section which includes the contents from Geopolitical and economic contexts of the Sahrawi refugee camp (page 3) to the end of this manuscript (page 9).

We have created a “Results” section in page 4 (including from page 3 to page 9, in previous text) and we have created numbered subheading on it (line 138 and subsequents).

Most importantly, the “Results” section should answer the research question, “Is it possible to become a nurse in a refugee camp?” (if this is the research question for this study).

Thank you very much form your comment. It is true. At the beginning of “Results” section we have added a paragraph which give the answer and briefly summarises its contents (lines 140-145).

Following Results or Finding section, the authors should create Discussion section that provides implications for practices and future research based on the authors’ findings.

We have added a “5. Discussion” section in page 10, which includes implications for practices and future research (lines 309-346). Also we have added new references (number 8, 9, 12, 13, 14, 19, 20, 21 and 22 in the new versión).

Moreover, right after the introduction section, there must be the section for “literature review” which provides an overview of Sahrawi refugee. Then, the authors should clarify how the authors’ study can fill the gap in the literature and what are the research questions for this study.

We have added “2. Literature review” section in page 3 (lines 71-112) providing an overview of Sahrawi refugees, and de gap existing in the literature.

Then, the authors should clarify how the authors’ study can fill the gap in the literature and what are the research questions for this study. Besides, the authors may need to consider the following points to enhance readability in the current form of the manuscript.

Abstract: “The only nursing school that exists worldwide in a refugee camp” should be edited. This is not a sentence, but must be a sentence. For example, “Sahrawi refugee camp is only the refugee camp worldwide where has nursing schools.”

Modified (line 29-30).

In page 2, please revisit and reword the sentence beginning “This paper is a result of the project of international development cooperation Contribution to the…” This sentence is too long, and it is not clear why the word “Contribution” began the upper case, “C.”

Modified: “contribution” with lower case, “c” (line 65). And we have divided this long sentence in two parts (line 66).

In page 3, under the section Geopolitical and economic context of the Sahrawi refugee camp, please clarify the meaning of “external cooperation.” It would be helpful to add some examples related to “external cooperation.”

We have added a concrete example to clarify the meaning of “external cooperation” (line 157).

In Page 3, there should be a connection between “Education in the Sahrawi camps” and “Ahmed Abdel-Fatah National School of Nursing:...” The nursing school might be established based on the two main goals that the authors described in the second paragraph of the section “Education in the Sahrawi camps.” If so, please add how the two main goals can be applied to the mission of the nursing schools.

We have added a little paragraph in page 5 to make the connection between both subheadings (lines 174-176).

We hope your agreement with changes and hearing soon from you.

Thank you.

Reviewer 2 Report

The article is a fine contribution to look into health aid in refugee study. It shows what nursing school is like and how could one be a nurse in the refugee camp.

however, the article is intrinsically problematic. It has not shown whether there are existing studies researching on this topic; how this article has contributed to the study; the methodology part and what the insight is. It is more merely more like a descriptive essay instead of a research article.

It will need a major buttering-up to be publishable.

Author Response

Cover letter – Reviewer 2

Comments to Suggestions

Thank you very much for the suggested modifications for the paper Is it possible to become a nurse in a refugee camp?. Having them already done, all the authors believe that the article has improved.

First of all, we reconfirm the author list and the corresponding affiliation of them.

Every modification done is described below in green colour. In the paper you can find them in green colour:

The article is a fine contribution to look into health aid in refugee study. It shows what nursing school is like and how could one be a nurse in the refugee camp.

However, the article is intrinsically problematic. It has not shown whether there are existing studies researching on this topic; how this article has contributed to the study; the methodology part and what the insight is. It is more merely more like a descriptive essay instead of a research article.

It will need a major buttering-up to be publishable.

Thank you very much for your suggestions. As you indicate, we have substantially modified the article. Besides, there is agreement with other reviewer´s suggestions. To that end we have completely remodeled the article. We have added:

- “2. Literature review” section (page 3)

- “3. Materials and methods” section (page 3)

- “4. Results” section (page 4)

- “5. Discussion” section (page 10) with implications for practice and futures researches

- several references (number 8, 9, 12, 13, 14, 19, 20, 21 and 22 in the new versión)

All this changes you can see in green colour in the remodeled versión of the article.

We hope your agreement and hearing soon from you.

Thank you.

Reviewer 3 Report

The text concerns an important topic. It is worth publishing work on this forgotten conflict and refugee camp

But unfortunately the presented article is not a scientific text. Is a note. If scientific papers will be more focus on topics between lines 191 and 200 (in the presented text) or in ex. will present a result of interviews with nurses after all mention courses, or with nurses working in the field there will certainly fulfill the task and we can consider this as a research article. I look forward to seeing you again with a appropriate text.

Author Response

Cover letter – Reviewer 3

Comments to Suggestions

Thank you very much for the suggested modifications for the paper Is it possible to become a nurse in a refugee camp?. Having them already done, all the authors believe that the article has improved.

First of all, we reconfirm the author list and the corresponding affiliation of them.

Every modification done is described below in green colour. In the paper you can find them in green colour:

The text concerns an important topic. It is worth publishing work on this forgotten conflict and refugee camp

But unfortunately the presented article is not a scientific text. Is a note. If scientific papers will be more focus on topics between lines 191 and 200 (in the presented text) or in ex. will present a result of interviews with nurses after all mention courses, or with nurses working in the field there will certainly fulfill the task and we can consider this as a research article. I look forward to seeing you again with a appropriate text.

Thank you very much for your suggestions. As you indicate, we have substantially modified the article. Besides, there is agreement with other reviewer´s suggestions. To that end we have completely remodeled the article. We have added:

- “2. Literature review” section (page 3)

- “3. Materials and methods” section (page 3)

- “4. Results” section (page 4)

- “5. Discussion” section (page 10) with implications for practice and futures researches

- several references (number 8, 9, 12, 13, 14, 19, 20, 21 and 22 in the new versión)

All this changes you can see in green colour in the remodeled versión of the article.

We hope your agreement and hearing soon from you.

Thank you.

Round 2

Reviewer 1 Report

Under 2. Literature Review (page 3), I feel the second and third paragraphs can be combined, which suggests the third paragraph beginning “This provoked the exodus…” can be located right after the last sentences of the second paragraph beginning, “Mauritania withdrew from the fights….” Literature Review (page 3): “Failing such agreement and due to the perpetuation of the situation of Sahrawi refugees…” can be edited like:

“Amid the deadlock of such political agreement, Sahrawi refugees’ harsh living conditions are perpetuated, which allows the government of the SADR to receive international aid [from where? UNHCR? Please note funding agencies] and develop the infrastructures necessary to provide basic healthcare.”

In this edited sentence, as I noted, please indicate the name of funding agencies where the government receive international aid.   

Literature Review (page 3): the sentence beginning “it was necessary to train young people…” is not clear to me. It may be edited like:

“…it was necessary to provide a nursing education program with young people who would later become professional nurses and help the refugees. The education program could not only fill the career and personal needs of the young people but also meet the health care needs of the refugees.”  

<Here, my question is: the young people could be refugee youth or other young people outside the refugee camps? If the young people include refugee youth, I would suggest the authors clearly indicate this. For example, …to provide a nursing education program with young people including refugee youth who would later…”>

Literature Review (page 3): the sentence beginning “This whole process, unique in the world…” may need to be edited. It may be edited like:

“This whole process, which spans from establishing a nursing school in a refugee camp to training young nursing professionals, is unique given that it has not been launched in any refugee camps across the world. Nevertheless, the extant literature has not documented this unique contexts yet.”

Material and methods (page 3): the sentence beginning “9 interviews of 45-60 minutes each…” may be edited like:

“9 interviews were conducted, each of which took approximately 45-60 minutes. However, the interviews were not recorded on audio or video (because the interviewees did not permit our recording --- if this is the case, I would suggest you indicate the reason the interviews were not recorded).

Materials and methods (page 4): the phrase beginning “selecting documents which provided..” should be transformed a sentence. For example, the authors may need to finish the previous sentence beginning “In the same way…” After this sentence, the authors may begin a new sentence like,

“The selected documents provided supplemental information, which can enrich the interview data.” ---> “the selected” may need to be changed “the obtained” because the authors obtained the documents rather than selected them.

Results (page 4): the sentence beginning “considering the perpetuity of such camps…” may need to be edited like

“Considering the perpetuity of such camps and the lack of healthcare professionals to serve the refugee population, building a nursing school inside the refugee camps is a priority for the Sahrawi government based on partnerships with a range of international organizations whose mission is to help refugees.”

In page 11 (Discussion), the authors need to put citation for the argument, “While, in 2016, 91% of children worldwide…84% non-refugees” In page 11 (Discussion), the authors may need to consider combining the quote of Malala Yousafzai and the last paragraph. So right after the quote, the authors may need to begin the sentence beginning “The geographical environment…”

Author Response

Major revision 2

Cover letter - Reviewere 1

Thank you very much for the new suggested modifications for the paper Is it possible to become a nurse in a refugee camp?

Every modification done is described below. In the paper you can find them in green colour:

Under 2. Literature Review (page 3), I feel the second and third paragraphs can be combined, which suggests the third paragraph beginning “This provoked the exodus…” can be located right after the last sentences of the second paragraph beginning, “Mauritania withdrew from the fights….”

Thank you very much for the suggestion. We have combined both paragraphs (lines 85-89).

Literature Review (page 3): “Failing such agreement and due to the perpetuation of the situation of Sahrawi refugees…” can be edited like:

 “Amid the deadlock of such political agreement, Sahrawi refugees’ harsh living conditions are perpetuated, which allows the government of the SADR to receive international aid [from where? UNHCR? Please note funding agencies] and develop the infrastructures necessary to provide basic healthcare.” In this edited sentence, as I noted, please indicate the name of funding agencies where the government receive international aid.  

We have edited this sentence and we have indicated the name of the funding agencies (lines 106-110).

Literature Review (page 3): the sentence beginning “it was necessary to train young people…” is not clear to me. It may be edited like:

“…it was necessary to provide a nursing education program with young people who would later become professional nurses and help the refugees. The education program could not only fill the career and personal needs of the young people but also meet the health care needs of the refugees.” 

We have edited this sentence as you suggested. Now, we think is clearer (lines 110-116).

<Here, my question is: the young people could be refugee youth or other young people outside the refugee camps? If the young people include refugee youth, I would suggest the authors clearly indicate this. For example, …to provide a nursing education program with young people including refugee youth who would later…”>

We have indicated that the young people were refugee youth (lines 111-112).

Literature Review (page 3): the sentence beginning “This whole process, unique in the world…” may need to be edited. It may be edited like:

“This whole process, which spans from establishing a nursing school in a refugee camp to training young nursing professionals, is unique given that it has not been launched in any refugee camps across the world. Nevertheless, the extant literature has not documented this unique contexts yet.”

Thank you very much for this suggestion. We have edited this sentence as you suggested (lines 114-116).

Material and methods (page 3): the sentence beginning “9 interviews of 45-60 minutes each…” may be edited like:

“9 interviews were conducted, each of which took approximately 45-60 minutes. However, the interviews were not recorded on audio or video (because the interviewees did not permit our recording --- if this is the case, I would suggest you indicate the reason the interviews were not recorded).

We have edited this sentence as you suggested (lines 128-130).

Materials and methods (page 4): the phrase beginning “selecting documents which provided..” should be transformed a sentence. For example, the authors may need to finish the previous sentence beginning “In the same way…” After this sentence, the authors may begin a new sentence like, “The selected documents provided supplemental information, which can enrich the interview data.” ---> “the selected” may need to be changed “the obtained” because the authors obtained the documents rather than selected them.

We have transformed this phrase and we have modified the indicated sentence (lines 142-144).

Results (page 4): the sentence beginning “considering the perpetuity of such camps…” may need to be edited like “Considering the perpetuity of such camps and the lack of healthcare professionals to serve the refugee population, building a nursing school inside the refugee camps is a priority for the Sahrawi government based on partnerships with a range of international organizations whose mission is to help refugees.”

Thank you very much for your suggestion. We have edited this sentence as you indicated. We think is clearer now (lines 151-155).

In page 11 (Discussion), the authors need to put citation for the argument, “While, in 2016, 91% of children worldwide…84% non-refugees”.

We have put he citation (line 334).

In page 11 (Discussion), the authors may need to consider combining the quote of Malala Yousafzai and the last paragraph. So right after the quote, the authors may need to begin the sentence beginning “The geographical environment…”.

We have combined the quote of Malala Yousafzai with the next paragraph (lines 353-356).

Reviewer 2 Report

The article is still problematic. I think the authors do not understand what literature review is. THey basically just gave me the background without discussing what are the existing literature about the subjects and how those studies' findings and how the current study is built upon those studies.

Author Response

Major revision 2

Cover letter - Reviewere 2

Thank you very much for the new suggested modifications for the paper Is it possible to become a nurse in a refugee camp?

Every modification done is described below. In the paper you can find them in green colour:

The article is still problematic. I think the authors do not understand what literature review is. THey basically just gave me the background without discussing what are the existing literature about the subjects and how those studies' findings and how the current study is built upon those studies.

Thank you very much for your suggested modification. We have added a paragraph at the end of section Literature review to highlight the non-existence of scientific articles addressing this topic, with the exception of some information found on different websites from NGDO´s that collaborated with the school of nursing, which have been duly cited. Thus, this article is intended to fill this blank in scientific literatura (lines 117-121).

Reviewer 3 Report

Thanks for a changes.

This time I would suggest to clear information in chapter 4.1 line146-154 that Sahrawi people are leaving in two territories....in fact three if we calculate refuge camps on Algerian land plus division of Western Sahara area.

Author Response

Major revision 2

Cover letter – Reviewere 3

Thank you very much for the new suggested modifications for the paper Is it possible to become a nurse in a refugee camp?

Every modification done is described below. In the paper you can find them in green colour:

Thanks for a changes.

This time I would suggest to clear information in chapter 4.1 line146-154 that Sahrawi people are leaving in two territories....in fact three if we calculate refuge camps on Algerian land plus division of Western Sahara area.

Thank you very much for your suggestion. We have made it clear that Sahrawi people live in three territories (lines 160-162).